# Interlaboratory Study of Ice Adhesion Using Different Techniques

**Sigrid Rønneberg [1]** **, Yizhi Zhuo [1], Caroline Laforte [2], Jianying He [1]** **and Zhiliang Zhang [1],***

1   Department of Structural Engineering, Norwegian University for Science and Technology (NTNU), NO-7491 Trondheim, Norway; sigrid.ronneberg@ntnu.no (S.R.); yizhi.zhuo@ntnu.no (Y.Z.); jianying.he@ntnu.no (J.H.)
2   Anti-Icing Materials International Laboratory (AMIL), Université du Québec à Chicoutimi, 555 Blvd. de l'Université, Chicoutimi, QC G7H 2B1, Canada; Caroline_Laforte@uqac.ca
*   Correspondence: zhiliang.zhang@ntnu.no; Tel.: +47-73592530 & +47-93001979

**Abstract:** Low ice adhesion surfaces are a promising anti-icing strategy. However, reported ice adhesion strengths cannot be directly compared between research groups. This study compares results obtained from testing the ice adhesion strength on two types of surfaces at two different laboratories, testing two different types of ice with different ice adhesion test methods at temperatures of −10 and −18 °C. One laboratory used the centrifuge adhesion test and tested precipitation ice and bulk water ice, while the other laboratory used a vertical shear test and tested only bulk water ice. The surfaces tested were bare aluminum and a commercial icephobic coating, with all samples prepared in the same manner. The results showed comparability in the general trends, surprisingly, with the greatest differences for bare aluminum surfaces at −10 °C. For bulk water ice, the vertical shear test resulted in systematically higher ice adhesion strength than the centrifugal adhesion test. The standard deviation depends on the surface type and seems to scale with the absolute value of the ice adhesion strength. The experiments capture the overall trends in which the ice adhesion strength surprisingly decreases from −10 to −18 °C for aluminum and is almost independent of temperature for a commercial icephobic coating. In addition, the study captures similar trends in the effect of ice type on ice adhesion strength as previously reported and substantiates that ice formation is a key parameter for ice adhesion mechanisms. Repeatability should be considered a key parameter in determining the ideal ice adhesion test method.

**Keywords:** ice adhesion; interlaboratory; ice removal; ice type; anti-icing; icephobic

## 1. Introduction

Anti-icing surfaces, or icephobic surfaces, are a promising technique for passive ice removal and may help mitigate and avoid dangerous situations and unwanted icing in our daily life [1–4]. The most promising strategy for anti-icing surfaces is low ice adhesion surfaces, where the ice automatically detaches from the surface by its own weight or natural forces [5–7]. However, although the amount of research on low ice adhesion surfaces has steadily increased over the past few years [8] and record low ice adhesion strengths of below 1 kPa have been reported [9–11], each research group develops its own custom-built set-up for measuring ice adhesion strength [9,12–15]. As a result, reported ice adhesion strength measurements cannot be directly compared [7,8,16,17].

In this experimental study, the research groups at the Anti-icing International Materials Laboratory (AMIL) at the University of Québec in Chicoutimi and the Nanomechanical Lab at the Norwegian University of Science and Technology (NTNU) collaborated to compare obtained ice adhesion strength measurements from two commonly available surfaces. Both have custom-built laboratory facilities able

to measure internally comparable ice adhesion strength in controlled environments. The ice adhesion strength was measured with a centrifuge adhesion test (CAT) at AMIL, and with a vertical shear test (VST) at NTNU. The centrifuge test is one of the most repeatable ice adhesion tests, although it cannot produce stress–strain curves [8,17,18]. For larger facilities, the CAT is a common way to measure ice adhesion strength, often for impact ice types produced with a freezing drizzle or in-flight icing simulation [19–31]. The VST is very common due to its simple and economical set-up and performance, although the location of the force probe impacts the ice adhesion strength greatly [32], and the stress distribution may not be completely uniform [8,17,18]. The VST is commonly in use by several research groups [7,11,32–39], and has been suggested as a standard for ice adhesion measurement utilizing only commercially available instruments [14].

When comparing reported ice adhesion strengths, it is also necessary to include the type of ice tested. Measured ice adhesion strength is highly dependent on the ice tested [40], and it is essential to test ice adhesion strength with a realistic ice type for low ice adhesion surfaces with a specific application in mind. In this study, both ice from freezing precipitation and ice from bulk water samples were tested (see Figure 1). These ice types are analogous to those presented elsewhere [40], and while precipitation ice (PI) is a form of ice from impacting freezing supercooled droplets (Figure 1a), bulk water ice (BWI) is a static, non-impact type of ice (Figure 1b,c). BWI is the most common ice for testing of ice adhesion strength [5,9,10,12,33,34,41–50], although PI has also been studied [19,24,51,52]. For most practical applications, PI is more realistic than BWI [8,17].

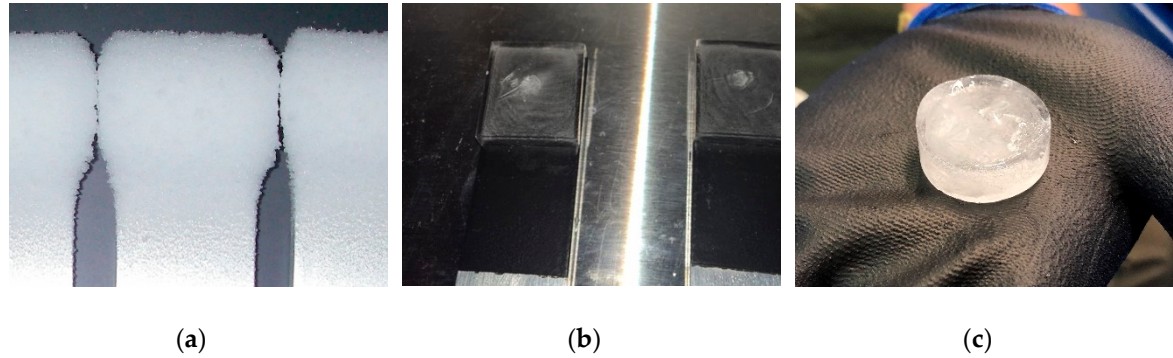

| **(a)** | **(b)** | **(c)** |

**Figure 1.** Pictures of the ice types tested in the study. (**a**) Illustration of precipitation ice (PI) created at the Anti-icing International Materials Laboratory (AMIL) (Tair = −18 °C); (**b**) illustration of bulk water ice (BWI) created at AMIL (Tair = −18 °C); (**c**) illustration of bulk water ice created at the Nanomechanical Lab at the Norwegian University of Science and Technology (NTNU) (Tair = −18 °C).

The comparison of ice adhesion strength measured at the AMIL and NTNU for the two types of ice showed that all results are comparable within the general trends between the NTNU and AMIL, with the greatest differences for bare aluminum surfaces at −10 °C. However, there are considerable differences between different laboratories. The study provides further evidence that ice formation is a key parameter in predicting the ice adhesion on different surfaces.

## 2. Materials and Methods

The ice adhesion strength of two surfaces were tested by both the AMIL and NTNU in their respective facilities. The surfaces tested were bare aluminum 6061-T6, and aluminum covered with EC-3100, a two component, water-based, icephobic, non-stick coating from Ecological Coating, LLC (New York, NY, USA) [53]. The testing of this icephobic coating has been reported previously [52,54]. The bare aluminum samples were polished with Walter BLENDEX Drum fine 0724 M4 (Windsor, CT, USA). To ensure similar surfaces, all the tested surfaces were prepared at AMIL facilities and transported to NTNU for testing. Each surface was tested only once to discount the durability aspect of the surfaces. All ice was generated with demineralized water of resistivity 18 MΩ cm. Both

temperatures of −10 °C and −18 °C were tested with six different samples from each configuration to generate average ice adhesion strength. Full experimental protocol is available as part of the Supplementary Materials.

### 2.1. AMIL Facility

The samples tested at AMIL were in the form of bars fit to the CAT apparatus, with the iced area on one side and a counterweight on the other. The bars had a length of 340 mm and thickness of 6.3 mm, with icing occurring over an area of about 1100 mm$^2$. This area was measured more precisely after the ice adhesion test in order to have the exact ice-surface detached surfaces.

PI was created through a freezing drizzle in a cold room of constant temperature and a relative humidity of 80% ± 2%. Six samples were iced simultaneously, with water of a median volume drop diameter (MVD) of 324 μm and an initial temperature of 4 °C at the exit of the sprayer nozzle. The surfaces had initial temperatures of the testing temperature, meaning either −10 °C or −18 °C. As the water hit the sample surface, it supercooled and froze on contact. Water impact speed is due to gravity as the water droplets fall from the nozzle; it is estimated to about 5 ms$^{-1}$. The samples were iced for 33 min and kept in a cold room for 1 h between icing and the ice adhesion test to allow the ice to thermally stabilize.

BWI was created in the same cold room by freezing water in silicon molds from MoldMax30 by Smooth-On (Macungie, PA, USA) [55]. The silicon molds had the same dimensions as the area iced during the freezing drizzle to generate ice samples as similar as possible to the PI. The molds were filled full of water, with the samples placed on top of the molds in contact with the water for freezing to occur. The surfaces and water were at room temperature at the start of the icing. Freezing time was 3 h, after which the molds were removed. The ice adhesion test was conducted after 15 min, in which the samples were weighed and measured.

The ice adhesion strength was measured with the CAT apparatus developed at AMIL [52] (see Figure 2). The CAT apparatus consists of a centrifuge, a placed sample beam, a counterweight to stabilize the bar with the ice sample, and a cover. The apparatus was placed within a cold room, ensuring in situ measurements of the ice adhesion strength for PI and BWI. The balanced and iced sample bars were spun in the centrifuge at an accelerating speed of 300 rpms$^{-1}$ until the ice was detached by the centrifugal force. Piezoelectric cells situated around the cover instantly detected the detachment of the ice, giving a detachment angular velocity. The ice adhesion strength was calculated as the centrifugal shear stress at the position of the center of mass of the ice sample at detachment divided by the ice-solid contact area [52].

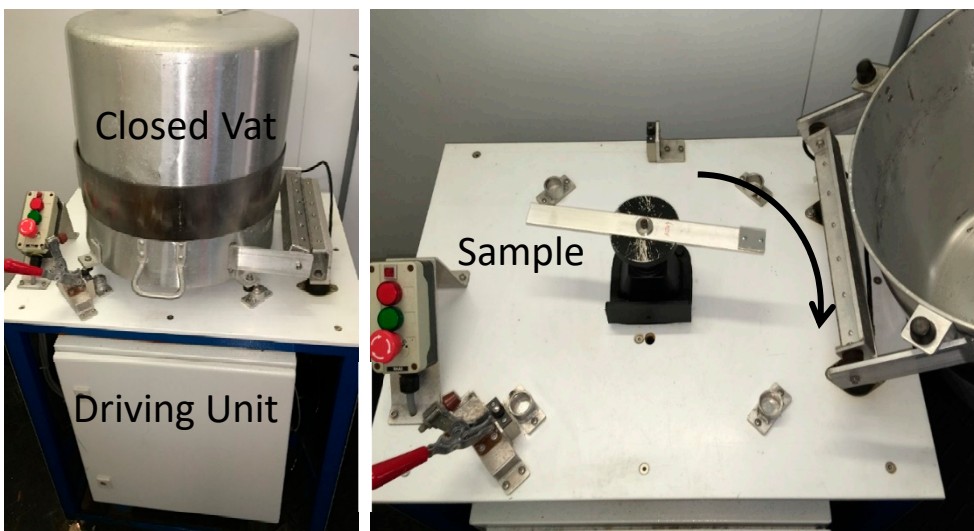

**Figure 2.** AMIL centrifuge adhesion test (CAT) apparatus.



### 2.2. NTNU Facility

The surfaces tested at NTNU were approximately square surfaces with a width of 7.3 cm, height of 7.2 cm, and thickness of 25 mm. The ice sample was frozen in the middle of the surface for testing. Both water and surfaces where initially at room temperature for the testing at NTNU.

The ice tested at NTNU was BWI. For the temperature of −18 °C, the ice samples were frozen in a freezer, while for the temperature of −10 °C, the ice was frozen in a cold room situated at a slight distance from the ice adhesion test. For both temperatures, the ice was frozen ex situ, and required transportation through room temperature to the testing rig where the samples were again placed in the original temperature for ice adhesion tests. For −18 °C, the transport time was about one min and 30 s, while for −10 °C, the transport time was about three min. To account for the transport from the cold room, the samples were transported in a box made of expanded polystyrene with freezer elements. Both the box and freezer elements were placed in the cold room for thermal equilibration before and after the transportation. After the transportation, the ice samples were placed in the ice adhesion test chamber for 15 min before testing to achieve thermal stability.

The BWI samples were frozen on the tested surfaces in a polypropylene centrifuge tube mold with a wall thickness of 1 mm and inner diameter of 27.5 mm. Silicone grease [56] was used to fasten the tube mold to the tested surface to avoid leakage during water insertion. Then, 5 mL of deionized water was inserted into the mold with a syringe to avoid air at the ice-solid interface, and pressure from a 200 g metal cylinder was placed on top of the tube to avoid water leakage during freezing. The water was frozen for 3 h before it was moved to the testing apparatus.

The ice adhesion test was performed with a VST and a custom-built set-up as modeled from other facilities [14] (see Figure 3). The detachment force was measured with an Instron machine (model 5944, Norwood, MA, USA) with load cell capacity of 2 kN (2530 Series static load cells), equipped with a home-built cooling system and chamber. The force probe fixed to the load cell was 5 mm in diameter and imposed an increasing force on the tube-encased ice samples with an impact velocity of 0.01 mms$^{-1}$. The placement of the probe was at the same point on the sample each test, situated 3 mm away from the tested surface during loading. The loading curve was recorded, and the peak value of the shear force was divided by the contact area to obtain the ice adhesion strength. As the probe distance is small and the measured ice adhesion strength is above 10 kPa for all tests, gravity can be discarded as negligible [8].

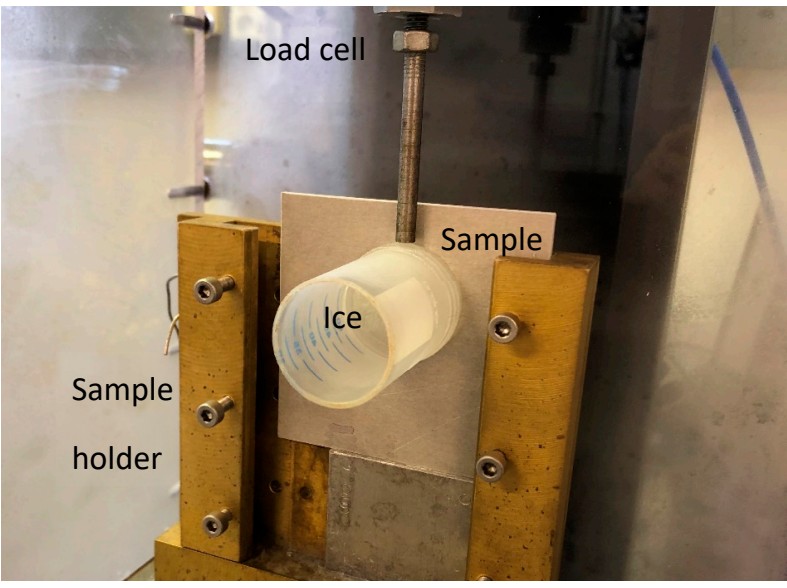

**Figure 3.** Vertical shear test (VST) as utilized at NTNU to measure ice adhesion strength.

## 3. Results

The measured ice adhesion strengths are shown in Figure 4. It can be seen that all results are comparable to a degree, with the greatest differences for bare aluminum surfaces at −10 °C. Table 1 presents an overview of all the ice adhesion strengths obtained from both laboratories. To obtain an average value, six different samples were tested at AMIL, except for BWI on bare aluminum at −10 °C where only four samples could be tested. At NTNU, averages were created from five samples. All the data are given in the supplementary materials.

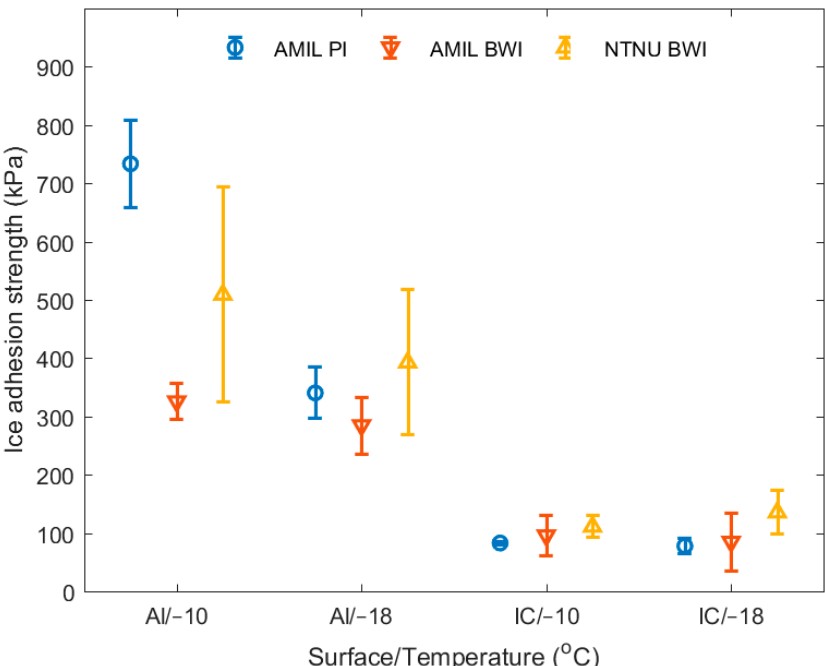

**Figure 4.** Measured ice adhesion strengths. Aluminum surfaces are denoted as Al, while the surfaces with icephobic coating are denoted as IC. All three ice types created are shown for each surface-temperature combination. Data displayed in Table 1.

**Table 1.** Overview of mean values and standard deviations of ice adhesion strength. Data illustrated in Figure 4, with all data available in the supplementary information.

| Surface/Temperature | Ice Adhesion Strength (kPa ± SD (%)) | | |
|---|---|---|---|
| | **AMIL PI** | **AMIL BWI** | **NTNU BWI** |
| Aluminum/−10 °C | 734 ± 75 (10%) | 326 ± 30 (9%) | 509 ± 185 (36%) |
| Aluminum/−18 °C | 340 ± 44 (13%) | 285 ± 49 (17%) | 393 ± 124 (32%) |
| Coating/−10 °C | 83 ± 3 (4%) | 96 ± 34 (35%) | 111 ± 19 (17%) |
| Coating/−18 °C | 78 ± 14 (18%) | 85 ± 49 (58%) | 135 ± 38 (28%) |

From Figure 4, it can be seen that for BWI, the NTNU VST method systematically yields higher ice adhesion strength than the AMIL CAT method for both aluminum surfaces and the icephobic coating. However, the standard deviation depends on the surface type. For bare aluminum, the deviation for VST is higher than CAT, while for the icephobic coating, the opposite trend is observed.

The original measurements for the ice adhesion tests are displayed in Figure 5 for the CAT and Figure 6 for the VST. For CAT at AMIL, the original measurements consisted of rounds per minute (RPM) vs. time, including the constantly increasing RPM in the centrifuge combined with the piezoelectric signal indicating the RPM at ice detachment. For VST at NTNU, the original measurements were of force per time.

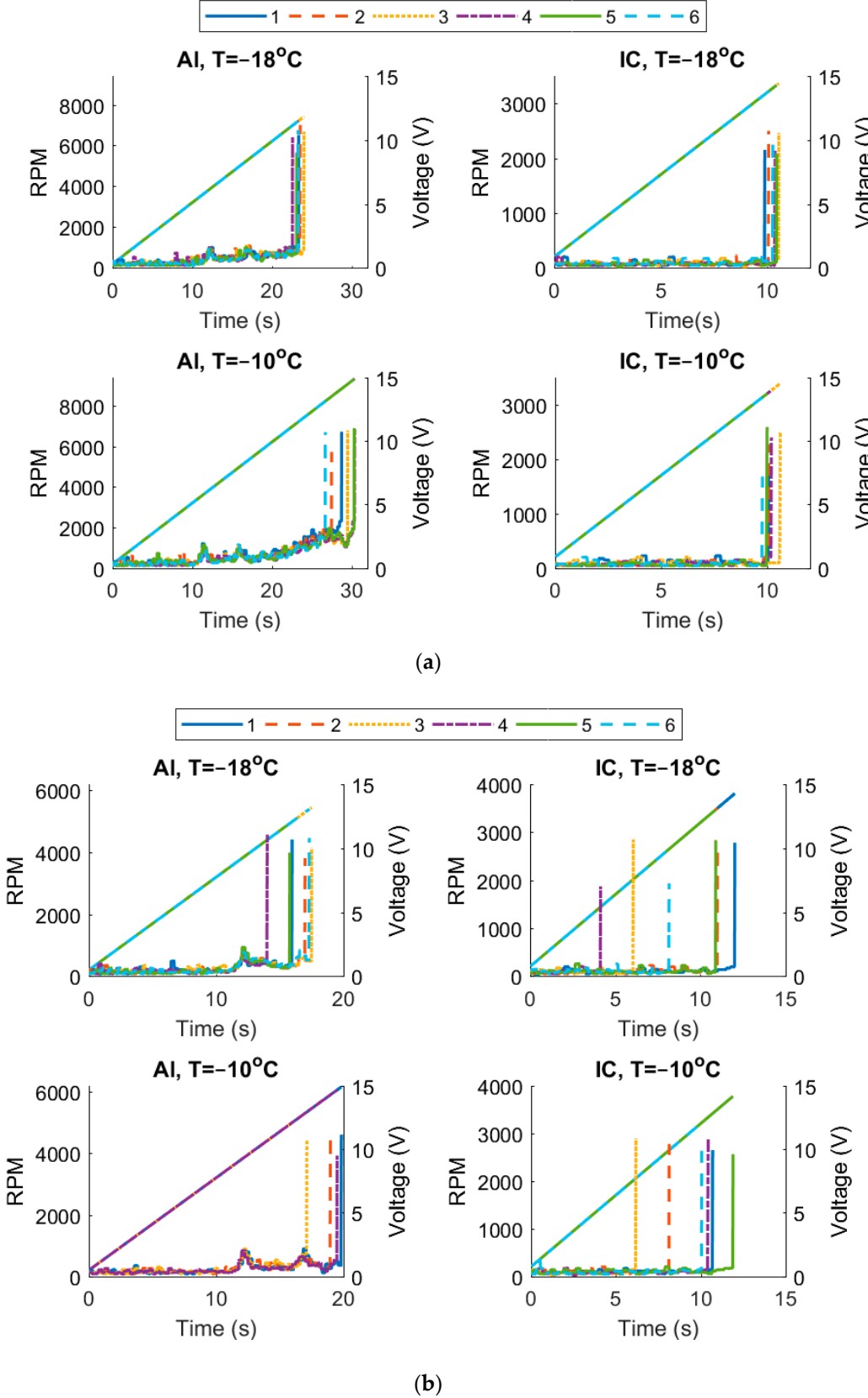

**Figure 5.** Rounds per minute (RPM)-time curves for CAT measured at AMIL for both ice types. The left axis denotes the RPM, which increases with constant acceleration, and the right axis displays the voltage measured by the piezoelectric cells. The end RPM was utilized to calculate the ice adhesion strength as described elsewhere [52], and can be deduced by the placement of the piezoelectric signal for each sample. All six samples for each surface and temperature are indicated, for (**a**) PI and (**b**) BWI.

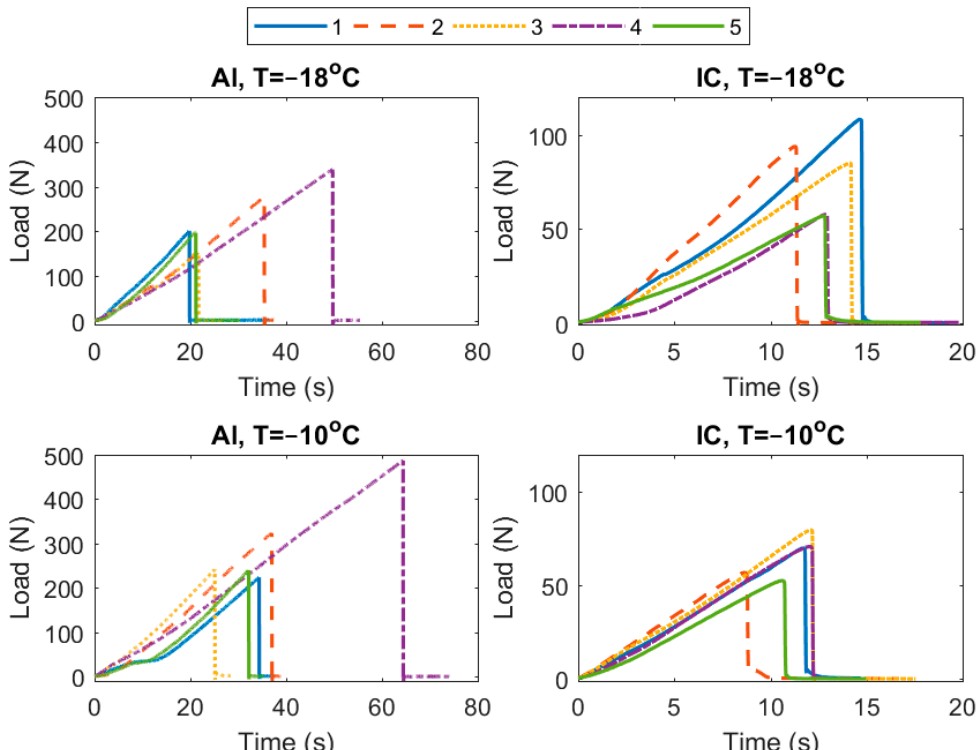

**Figure 6.** Force–time curves for VST ice adhesion measurements performed at NTNU. The maximum force was divided by the contact area of the ice sample to calculate the ice adhesion strength. All five samples for each surface and temperature are indicated.

## 4. Discussion

When comparing the two surface types for all ice types, all tests showed larger error bars for aluminum than for the icephobic coating. This high standard deviation is in accordance with other studies of ice adhesion strength and may be an inherent property of the ice removal mechanisms [8,56]. The ice adhesion strengths for the icephobic coating from both laboratories are close to each other, but show larger variations for BWI up to 58%, compared to only 18% for PI.

The fluctuating standard deviations can be partially explained by the original measurements in Figures 5 and 6. In Table 1, the instances with a standard deviation above 30% are bare aluminum surfaces tested at NTNU and the icephobic coating with bulk water ice at AMIL. For all these instances, it can be seen in the original measurements that there are significant outliers. In other words, some samples for these cases display significantly changing behavior concerning ice adhesion strength, which greatly impacts the mean value and standard deviation. The standard deviation for the icephobic coating is generally lower than for bare aluminum, because there are fewer outlines in these measurements, with the exception of bulk water ice on icephobic coating tested at AMIL in Figure 5b. The standard deviations should be investigated further by expanding on this interlaboratory study with more samples and more tests, in addition to including more laboratory facilities.

The most commonly utilized configuration when testing ice adhesion strength is a shear test analogous to VST with bulk water ice at −10 °C [8]. As seen in Figure 6, this configuration includes a high standard deviation for bare aluminum and is where the greatest difference between the two laboratories and ice types are found. However, when the outlier from NTNU is removed, the mean ice adhesion strength becomes 441 MPa, with a standard deviation of 17%. These values are much closer to what would be expected based on the rest of the tests and configurations. However, a goal of this interlaboratory study was to perform the tests in the default manner of the two laboratories. Although the outlier for bare aluminum at −10 °C at NTNU displayed partly cohesive failure, as seen when comparing Figures S10 and S12 in the supplementary materials, the failure was not clearly cohesive

and as such would not have been excluded from the study. If it is assumed that the tendency for outliers is possible for all ice adhesion tests, it might indicate why the standard deviation is generally high for ice adhesion measurements for all test methods.

The effect of decreasing temperature varied for the tested surfaces. At AMIL, there was a marked decrease of ice adhesion strength for PI on bare aluminum, and a lesser decrease for the icephobic coating as well. This decrease is due to the increased occurrence of cohesive failures. Between −10 and −18 °C, there is a transition from adhesive failures to more cohesive failures for aluminum and PI, as shown previously [24]. The same transition can be seen for the icephobic coating (see supplementary materials). At NTNU on the other hand, there was only one occurrence of partial cohesive failure for bare aluminum surfaces, which occurred at −10 °C when using the VST. These observations indicate that the transition to cohesive failures and the occurrence and impact of non-adhesive failures depends on the ice adhesion test method and ice type. Furthermore, the occurrence of cohesive failure displays no relation to the standard deviations in Table 1 and outliers in Figures 5 and 6. For the BWI on the icephobic coating tested at NTNU, there is a slight increase of ice adhesion strength with temperature. The varying effect of temperature on the ice adhesion strength for the different configurations substantiate the difficulty in predicting the dependence of ice adhesion strength on temperature, as reported previously [17].

In general terms, this study shows that there are large differences between different laboratories, and that the differences do not seem to be systematic. It seems that for higher ice adhesion strengths, the difference between different ice adhesion tests and ice types increases. It follows that more tests with a larger range of ice adhesion values are needed to explore this relation more fully.

As two different ice types were tested at AMIL, a similar trend from Rønneberg et al. [40] can be seen in that BWI has a lower ice adhesion strength than PI for bare aluminum. However, for the icephobic coating, the ice adhesion strength for both ice types is very similar. As a result, it may be that the difference in ice adhesion strength between different types of ice depends on whether the tested surface is defined as a low adhesion surface.

When comparing the results from AMIL and NTNU, some general comments about different ice adhesion measurement set-ups can be made. At low ice adhesion, the two test methods gave similar results, while the VST seemed to give larger deviations than the CAT methodology. However, the VST was easier to implement, and had a slightly lower standard deviation for low ice adhesion surfaces with BWI. An alternative might be the lap shear test, as studied recently [57], although no comparison can be made between this new test method and the ones presented in this study. As the outliers that differed in ice adhesion strength from their peers greatly impacted the standard deviation for the ice adhesion strength tests, repeatability should be a key factor in determining the ideal ice adhesion test.

Lastly, some additional sources of error present in the experiments reported here must be mentioned. For the tests performed at NTNU, the ice adhesion tests were performed ex situ and the ice samples and tested surfaces were moved between the freezer to the testing apparatus. Especially the tests performed at −10 °C were subject to a long transport between two different laboratories, and to account for this thermal variation, a polystyrene container was used. The effect of this container compared to the shorter transport at room temperature for the tests performed at −18 °C cannot be determined exactly. However, despite the transport which was assumed detrimental for ice adhesion, the NTNU VST method yielded higher ice adhesion for both coatings, compared to AMIL results where the experiments were performed in situ. This observation may indicate that the transportation did not significantly affect the ice adhesion.

The ice sample size differed between AMIL and NTNU, with an iced area of about 1100 mm$^2$ at AMIL and only 594 mm$^2$ at NTNU. While at AMIL, the ice samples covered the entire tested surface as seen in Figure 1a,b; at NTNU the ice sample was situated at a part of the tested surface only, as seen in Figure 3. The fact that the ice sample at NTNU was smaller compared to the surface structure, especially for aluminum, may be a factor in the much higher standard deviation seen for the aluminum samples from NTNU than the icephobic coating.

## 5. Conclusions

In this study, the ice adhesion strength of two different surfaces were tested at two laboratories with different ice adhesion test methods and two types of accreted ice. Despite the differences between the laboratories, the experiments capture the overall trends in which the ice adhesion strength surprisingly decreased from −10 °C to −18 °C for bare aluminum and was almost independent of temperature for a commercial icephobic coating. For BWI, the NTNU VST method systematically yielded higher ice adhesion strength than the AMIL CAT method. The standard deviations were approximately constant when testing PI at AMIL and seem to scale with the absolute value of ice adhesion at NTNU. The VST had higher deviations than CAT methodology for high ice adhesion values but were more similar when testing low ice adhesion surfaces. For configurations with standard deviation above 30%, the ice adhesion tests showed more significant outliers that differed from the other tests for that configuration than those with a smaller standard deviation.

The experiments in this study were performed with a focus on keeping the conditions similar, both within each lab and between AMIL and NTNU. However, the results still show significant differences and variations for all configurations. As a result, more data from several more laboratory facilities are needed, as well as more tests within each laboratory facility. Furthermore, the study provides further evidence that the ice formation is a key parameter in predicting the ice adhesion on different surfaces, as well as for the investigation of the mechanism of the ice detachment from different surfaces and the occurrence of cohesive failures during ice adhesion testing. To determine the ideal ice adhesion strength test, repeatability is a key factor to minimize the number of experimental outliers which greatly impact the standard deviation.

**Supplementary Materials:** The following are available online at http://www.mdpi.com/2079-6412/9/10/678/s1; Table S1: Experimental protocol, Table S2: All experimental results, Figure S1: Formation BWI at AMIL, Figure S2 and S3: Formation BWI at NTNU, Figure S4 and S5: Typical adhesive failure at AMIL, Figure S6–S9: Typical cohesive failure at AMIL, Figure S10 and S11: Typical adhesive failure at NTNU, Figure S12: Cohesive failure at NTNU, Figure S13: Adhesion reduction factor (ARF).

**Author Contributions:** All authors contributed in conceptualization, methodology and writing—review and editing; investigation S.R., Y.Z., and C.L.; visualization, formal analysis, and writing—original draft, S.R.; supervision and project administration, C.L., J.H., and Z.Z.; funding acquisition, J.H. and Z.Z.

**Funding:** The authors gratefully acknowledge the financial support from the Norwegian Research Council FRINATEK project Towards Design of Super-Low Ice Adhesion Surfaces (SLICE, 250990) and from the PETROMAKS2 project Durable Arctic Icephobic Materials (AIM, 255507).

**Conflicts of Interest:** The authors declare no conflicts of interest.

## Nomenclature

| | |
|---|---|
| AMIL | Anti-icing Materials International Laboratory |
| ARF | Adhesion reduction factor |
| BWI | Bulk water ice |
| CAT | Centrifuge adhesion test |
| F | Centrifugal force |
| IC | Icephobic coating |
| MVD | Median volume drop diameter |
| NTNU | Norwegian University of Science and Technology |
| PI | Precipitation ice |
| RPM | Rounds per minute |
| VST | Vertical shear test |

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
