# Peer review of "Interlaboratory Study of Ice Adhesion Using Different Techniques"

_coatings, doi:10.3390/coatings9100678_

Round 1

Reviewer 1 Report

 Although the title of this paper was promising and potentially interesting, I have to admit that I increasingly lose my interest while reading.

I was expecting a wider comparison between samples and techniques, but I found that the paper only compares the shear strength results provided by only two techniques and two samples. These two samples are bare aluminum and aluminum coated with a commercial coating, which details about its composition are missing. Other samples (samples fabricated by the authors, for instance) would be more recommendable to increase the quality and attractiveness of the work.

Authors only reported, but they did not provide any hypothesis or explanation why the results agree sometimes, and sometimes not. They used a different strategy to produce ice and they claim that how the ice is produced, seems to be a key factor. However, one finds that with the exception of the bare aluminum sample for a specific temperature (T=-10ºC), the use of precipitation ice or bulk water ice does not really play a significant role.

All the discussion of this paper is based on a single Figure and a single Table that shows exactly the same results that the Figure, but displayed in a different manner.

In my opinion, this paper does not deserves publication unless the authors include more samples, more techniques and more discussion. My impression is that this paper needs much more work to be recommendable for publication.

A minor point:

Within the introduction section, authors did not delete the text that was included within the “Guide for authors” or template (lines 52-59).

Author Response

Point-by-point response - Reviewer 1

Manuscript ID: coatings-605216

Although the title of this paper was promising and potentially interesting, I have to admit that I increasingly lose my interest while reading.

We thank the reviewer for the interest in what we aim to communicate. In order to make the limitations of the study even clearer, we have added more moderators in the introduction. These moderations will enable the reader to more quickly grasp the core and uncertainties of the study.

I was expecting a wider comparison between samples and techniques, but I found that the paper only compares the shear strength results provided by only two techniques and two samples. These two samples are bare aluminum and aluminum coated with a commercial coating, which details about its composition are missing. Other samples (samples fabricated by the authors, for instance) would be more recommendable to increase the quality and attractiveness of the work.

We considered applying the coatings developed at the lab at NTNU for this interlaboratory study, for example the coating as reported by a recent paper published in Coatings (Li et al,  https://doi.org/10.3390/coatings9100602). However, to use such innovative coatings would remove the element of comparison from the study. The idea is that to use only these two surfaces, which are commercially available and easy to obtain for all laboratory facilities, this study might be reproduced in other research groups as well. The goal is not to discuss the icephobicity of the surfaces tested, but rather to discuss the ice adhesion strength of the same surfaces, tested with different ice adhesion tests and ice types.

The reason only two laboratories were included in this study, is that the research was contrived in cooperation between these two institutions. We are currently in dialogue with several more facilities to extend the principles of the study reported here to a wider range of comparisons. However, a more comprehensive study requires more time, and less certainty in the constancy of parameters. With only two laboratories involved, we were able to control all parameters to the best of our ability, and agree on similar experimental procedures to ensure maximum comparison within the different equipment and experimental set-ups.

Concerning the icephobic coating, we have added more information about its origin and added a reference where a similar coating can be purchased commercially.

Authors only reported, but they did not provide any hypothesis or explanation why the results agree sometimes, and sometimes not. They used a different strategy to produce ice and they claim that how the ice is produced, seems to be a key factor. However, one finds that with the exception of the bare aluminum sample for a specific temperature (T=-10ºC), the use of precipitation ice or bulk water ice does not really play a significant role.

This study is a first attempt at comparability between laboratories. As the experiments we did were fairly simple and did not include investigation of the ice detachment, microstructure or other properties of the ice, nor a variation of parameters, we do not attempt to give a hypothesis or explanation of the results. Our aim is to report whether we see grounds for comparison or not, which we do to a degree. We also see differences between the ice types tested at AMIL, confirming earlier studies, with smaller differences between the tests with bulk water ice at different facilities. As a consequence, the type of ice, whether precipitation ice or bulk water ice, is deemed important by us for further comparison of ice adhesion strengths. Concerning the significance of the differences, due to the high standard deviation the data is not always significantly different from each other. However, for practical use in determining what results can and cannot be compared, we believe it is important to report the trends.

When it comes to bare aluminum samples for -10oC, we have in the revised manuscript expanded on the discussion for this configuration. This expansion is importance with respect to the comparability issue, as a large amount of the low ice adhesion research is performed with bulk water ice at -10oC with aluminum surfaces as reference. As a consequence, the high variability and the apparent outlier recorded at NTNU is important to be aware of.

All the discussion of this paper is based on a single Figure and a single Table that shows exactly the same results that the Figure, but displayed in a different manner.

Another reviewer requested more data, and as a consequence we have included the original measurements in RPM vs time curves from AMIL and force vs time curves from NTNU. This inclusion offers a greater range of discussion, and comparison. For instance, we have included a discussion from line 192 to 202 on the single outliers in the results, which impacts the standard deviation greatly. These outliers enables us to make some further comments on the ice adhesion test methods, such as the importance of repeatability to enable correct standard deviations.

Furthermore, we have stressed that all experimental data are available in the supplementary materials, so enable a full statistical discussion on the results presented in this study.

In my opinion, this paper does not deserves publication unless the authors include more samples, more techniques and more discussion. My impression is that this paper needs much more work to be recommendable for publication.

We agree with the reviewer that the ideal interlaboratory study includes more surfaces, more ice adhesion tests and more types of ice, both of similar ice types, such as different configurations of glaze ice, and very different ice types, such as impact ice vs non-impact ice. However, such a study is challenging to conduct, as all laboratories have their own opinions on what is the ideal procedure to measure ice adhesion. To be able to further the grounds for comparison, it is necessary to start with slightly lower expectations. We hope that this study with only two laboratories can be applied to further comparison with other facilities and research groups, as we have described our experimental procedure to ensure reproducibility. Instead of a very large group cooperating with each other, other research groups have the opportunity to compare their results with the results we describe in this manuscript.

We hope that, with the mentioned extension of the discussion and the inclusion of the original measurements, the reviewer is more inclined to recommend publication of the manuscript.

A minor point:

Within the introduction section, authors did not delete the text that was included within the “Guide for authors” or template (lines 52-59).

We thank the reviewer for noticing this mistake in our preparation of the manuscript. The irrelevant lines have been removed in production.

Reviewer 2 Report

It would be nice to have some more data to make a stronger statistical case.

It would be nice if some of the original measurements (stress vs time , rpm vs time) was shared in the paper to actually show how the adhesion strength was measured. Any similarities differences of mode of failure.

Also for the conclusions a more detailed discussion as to the variation of standard deviations. The sample size is small but it seems to me that for all test with coating they were significantly narrower than without. Any thoughts on that can this be traced back to physics of adhesion.. some more insight and may be recommendations to improve testing..

Author Response

We thank the reviewer for this suggestion to add more data. We have made it clearer that all the experimental data and results are available in the supplementary materials. Furthermore, we have added the original measurements as requested, see Figures 5 and 6. Furthermore, we have discussed the original measurements with respect to the standard deviations and identified outliers in the measurements. This discussion can be seen in lines 191-212, and includes a discussion where the largest outlier for the aluminum surface at -10oC tested at NTNU was removed.

The physics of adhesion is, in our opinion, outside the scope of this study. This study aims to be a report of an interlaboratory study designed to investigate the comparison of similar experiments performed at different laboratory facilities, and to discuss issues in comparability with the two ice adhesion tests applied. However, with the inclusion of the original measurements and an expansion of the discussion in the study, we include that repeatability should be a key factor in discussing the ideal ice adhesion test due to the large effect seen by a few outliers in the results (see line 202-212 in particular). Based on this discussion, we include a recommendation to pay close attention to both outliers in results, and repeatability, when discussion comparability of ice adhesion strength tests. How this discussion is related to the physics of adhesion is extremely interesting, and something that should be included in a larger study where the detachment mechanisms of ice during ice adhesion tests with different methods is included. Until then, we can do nothing but speculate. Hopefully, this aspect may be included in a future, more extended, interlaboratory study with more participants and resources.

Reviewer 3 Report

It is true that the test results of ice adhesion from different research groups cannot be directly compared. As discussed in the manuscript, it is not only because different methods are used, but also because of different ice formations. The work is useful to the experimental research and development of low ice adhesion surfaces. 

Author Response

We thank the reviewer for the comments, and appreciate and agree with the viewpoints expressed in the review.

Round 2

Reviewer 1 Report

I am still not fully satisfied with the changes made on the manuscript and the explanations given by the authors. I do not feel that this work deserves publication.